# Exact Two-Dimensional Analytical Calculations for Magnetic Field, Electromagnetic Torque, UMF, Back-EMF, and Inductance of Outer Rotor Surface Inset Permanent Magnet Machines

**AmirAbbas Vahaj \***[ID]**, Akbar Rahideh**[ID]**, Hossein Moayed-Jahromi and AliReza Ghaffari**

Department of Electrical and Electronics Engineering, Shiraz University of Technology, Shiraz 13876-71557, Iran; rahide@sutech.ac.ir (A.R.); h_moaied@yahoo.com (H.M.-J.); a.ghaffari@sutech.ac.ir (A.G.)

**\*** Correspondence: a.vahaj@sutech.ac.ir

**Abstract:** This paper presents a two-dimensional analytical model of outer rotor permanent magnet machines equipped with surface inset permanent magnets. To obtain the analytical model, the whole model is divided into the sub-domains, according to the magnetic properties and geometries. Maxwell equations in each sub-domain are expressed and analytically solved. By using the boundary/interface conditions between adjacent sub-regions, integral coefficients in the general solutions are obtained. At the end, the analytically calculated results of the air-gap magnetic flux density, electromagnetic torque, unbalanced magnetic force (UMF), back-electromotive force (EMF) and inductances are verified by comparing them with those obtained from finite element method (FEM). One of the merits of this method in comparison with the numerical model is the capability of rapid calculation with the highest precision, which made it suitable for optimization problems.

**Keywords:** analytical model; partial differential equations; separation of variable technique; electrical machines; surface inset permanent magnet

---

## 1. Introduction

The existence of different types of PM brushless machines (PMBLM) made them applicable for a wide range of applications. PMBLMs have superiorities in comparison with other rivals like induction machines or reluctance synchronous machines due to higher efficiency, high torque per volume, lower torque ripple, lower vibration, and lower acoustic noise.

PMBLM can be categorized in terms of various criteria such as the topology of PMs, the relative position of the rotor and stator, the slotted or slotless stator structure, etc.

Various PM topologies such as surface-mounted, surface-inset, and interior are used where each has its own advantages and disadvantages. Among these topologies, surface-inset can provide a compromise between the other two topologies.

Electric machines with single rotor and single stator can be either inner rotor or outer rotor. The outer rotor motors can develop more output torque than the inner ones for the same volume of the machine. Usually, inner rotor machines are used for applications, which need rapid acceleration and deceleration. Outer rotor machines usually are used for applications which need constant speed. Also, the mechanical robustness of the PMs in the outer rotor configuration is higher than the inner one.

In this paper because of the aforementioned advantages of the outer rotor machines and surface inset PMs, an exact two-dimensional electromagnetic model for this type of machines is extracted.

In the design procedure, the static model is normally considered. Numerous static models have been presented for electric machines, in which some of them are based on the analytical approaches [1–52], and the others are based on numerical methods [53–55].

The presented analytical model for electric machines are based on permeance model [1] or magnetic equivalent circuit, also known as (a.k.a.) 0-D analytic model [2], or resolution of the Maxwell's equations in 2-D plane (a.k.a. 2-D analytic model) [3–48] or 3-D plane (a.k.a. 3-D analytic model) [49,50]. Also, other methods based on mapping techniques such as Schwarz-Christoffel have been used to extract the model of some machine with the analytical approach [51,52]. The most accurate presented models among the analytical methods are 3-D analytic and 2-D analytic. The 2-D analytic method can be used instead of 3-D when the model is symmetric and has no skewing. Two-dimensional analytical models are not only capable of considering a high number of harmonics which elevate the precision of the models, but also have less computational time in comparison with the numerical methods and made them appropriate for optimal design problems.

Two-dimensional analytical models are presented for the slotted [3–17,19,20,29–38,41,42,44,45] and slotless [21–28] machines, equipped with surface mounted [3–6,10–12,14–20,23,26,32,34,37,41,52] and surface inset [9,13,21,22,24,25,35,36] or spoke type magnets [7,8], in order to obtain the important quantities like magnetic flux density, electromagnetic torque, unbalanced magnetic force, back-EMF, and inductances. Also, the 2-D analytical model is used to calculate the eddy current effect [11,18,22,31,32,49,50] in electrical machines. Most of the abovementioned publications are focused on the inner rotor structure [5–18,22–27,31–38,42,44–46] and only a few of them present the 2-D model for outer rotor machines with surface mounted PMs [4,28–30,41]. Therefore, to the best knowledge of the authors, it is for the first time that 2-D analytical model of brushless PM machines with outer rotor and surface inset PM is presented using the subdomain technique.

Most of the developed 2-D or 3-D analytic model are assumed with the infinite permeability of the iron parts. New techniques to account for finite soft-magnetic permeability have been recently developed, i.e., the multi-layer model using the Cauchy's product theorem is presented in [38], and the subdomain technique by applying the superposition principle in both directions is proposed in [39–46] which can be used to calculate the core losses and saturation phenomena.

An overview of the analytical models in the Maxwell-Fourier method with a global or local saturation effect has been realized in [40]. According to [48], Dubas' superposition technique [39,40] is very interesting since it enables the magnetic field calculation in the material of slotted geometries. This superposition technique has been implemented in radial-flux electrical machines with(out) PMs supplied by a direct or alternate current [44,45].

The presented technique in [39–46] is not only used to predict the magnetic field in all parts of the electrical machines, but also it is used to obtain a 2-D analytical model of the steady state heat transfer of the electrical machines by solving the heat equations [47].

The aim of this paper is to extract a 2-D analytical model of PM brushless outer rotor machines equipped with surface inset PMs. The model is used to analytically compute the electromagnetic torque, torque ripple, back-electromotive force (EMF), inductances and unbalanced magnetic forces (UMF).

Therefore, this paper is organized as follows. In Section 2 the procedure of extracting the 2-D model is explained. Section 3 is dedicated to the calculation of the electromagnetic quantities. In Section 4 the analytical results of the case study are presented and compared with those of the numerical method. In the final part this paper is concluded.

## 2. Extracting the Magnetic Model

### 2.1. Assumptions

Figure 1 shows the topology of an outer rotor surface inset brushless permanent magnet machine.

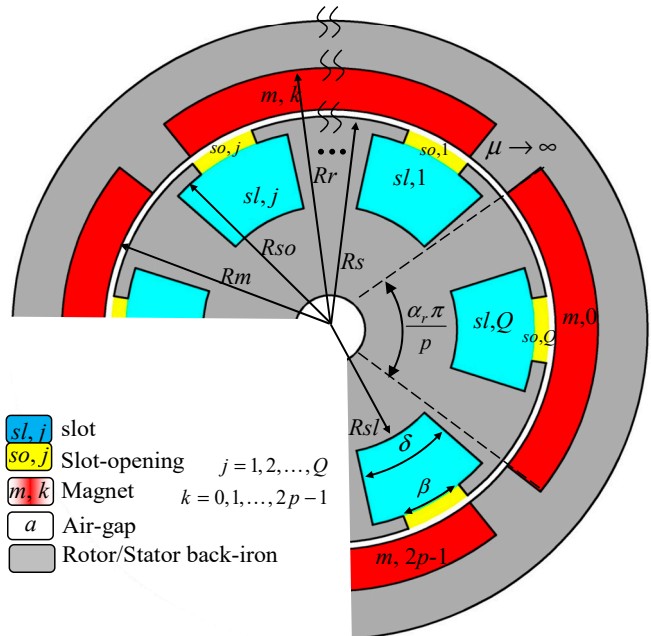

**Figure 1.** Outer rotor surface inset brushless permanent magnet machine illustration.

In order to make the problem solvable, below assumptions are made:

(a) According to the geometry and the absence of skewing, the problem is solved in 2-D polar coordinates which means the end effect is neglected.

(b) Magnetic vector potential has just axial component which is function of $r$ and $\theta$. Consequently, magnetic flux density has radial and tangential component; i.e., $\mathbf{A} = [0, 0, A_z]$, $\mathbf{B} = [B_r, B_\theta, 0]$.

(c) All materials are isotropic.

(d) Rotor and stator back iron have infinite permeability.

(e) The edges of the slots and slot-openings have radial direction.

(f) The eddy current effect is neglected.

### 2.2. Dividing Region into Sub-Regions

According to the shape and material characteristics, the whole domain is divided into a number of sub-domains. All the sub-domains are illustrated in Figure 1 and listed in Table 1 for a PMBLM with $Q$ slots and $p$ pole-pairs.

When the winding is single-layer or double-layer non-overlapping, as shown respectively in Figure 2a,b, each slot is a single subdomain; however, if the winding is two-layer overlapping, as shown in Figure 2c, each slot is divided into two subdomains.

**Table 1.** The sub-domains and related symbols.

| Sub-Domain | Symbol | Number of Sub-Regions |
|---|---|---|
| Magnet | $m$ | $1, 2, \ldots, 2p$ |
| Air-gap | $a$ | $1$ |
| Slot-opening | $so$ | $1, 2, \ldots, Q$ |
| Slot | $sl$ | $1, 2, \ldots, Q$ |

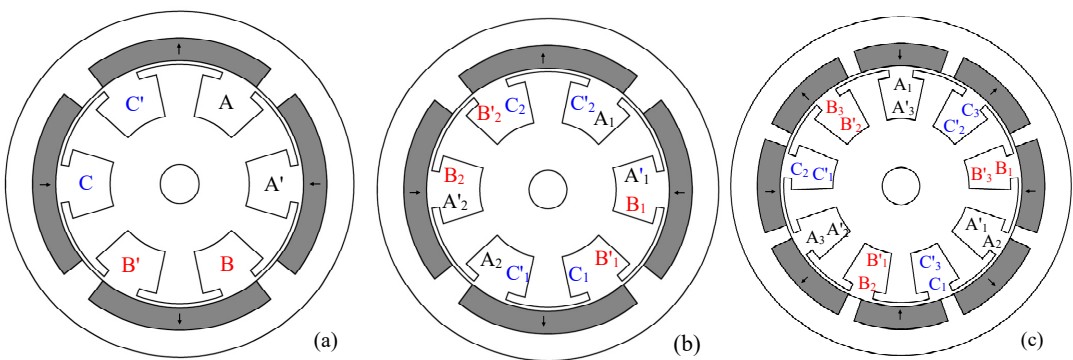

**Figure 2.** Winding topologies (**a**) single layer alternate teeth wound (**b**) double layer all teeth wound non-overlapping (**c**) double-layer overlapping.

### 2.3. Extracting the Magnetic Model

In this part, for each sub-domain a partial differential equation is extracted based on Maxwell's equations.

Maxwell's equations in quasi-static form are as follows:

$$\nabla \cdot \mathbf{B} = 0 \tag{1}$$

$$\nabla \times \mathbf{H} = \mathbf{J} + \hat{\mathbf{J}} \tag{2}$$

where **B** is the magnetic flux density vector. Ampere's law represents the relation between the magnetic field intensity vector (**H**), external current density vector (**J**), and current density vector in media ($\hat{\mathbf{J}}$). In this investigation the current density vector in media is assumed to be negligible, i.e., $\hat{\mathbf{J}} = 0$.

The relation between the magnetic flux density vector and the magnetic field intensity vector in permanent magnet media with linear demagnetizing curve is as follows:

$$\mathbf{B} = \mu_0 \mu_r \mathbf{H} + \mu_0 \mathbf{M} \tag{3}$$

where **M** is the magnetization vector.

Substituting (3) in (2) yields

$$\nabla \times \mathbf{B} = \mu_0 \mu_r \mathbf{J} + \mu_0 \nabla \times \mathbf{M} \tag{4}$$

The magnetic flux density vector can be represented as the curl of the magnetic vector potential (**A**):

$$\mathbf{B} = \nabla \times \mathbf{A} \tag{5}$$

Using (4) and (5) the following expression is obtained:

$$\nabla^2 \mathbf{A} = -\mu_0 \mu_r \mathbf{J} - \mu_0 \nabla \times \mathbf{M} \tag{6}$$

For each sub-domain, Equation (6) results in Poisson equations for the magnet and slot regions, and Laplace equations for the air-gap and slot-opening sub-domains, as represented below.

$$\nabla^2 \mathbf{A}^{sl,j} = -\mu_0 \mu_r \mathbf{J} \tag{7}$$

$$\nabla^2 \mathbf{A}^{m,k} = -\mu_0 \nabla \times \mathbf{M} \tag{8}$$

$$\nabla^2 \mathbf{A}^{\chi} = 0, \ \chi = \{(a), (so, j)\} \tag{9}$$

In 2-D polar coordinates, the magnetic vector potential and the current density vector have just a component along $z$, i.e., $\mathbf{A} = [0, 0, A_z(r, \theta)]$ and $\mathbf{J} = [0, 0, J_z(\theta, t)]$. Also, the magnetic flux density vector and the magnetization vector have radial and tangential components as below:

$$\mathbf{B} = [B_r(r, \theta), B_\theta(r, \theta), 0]$$

$$\mathbf{M} = [M_r(r, \theta), M_\theta(r, \theta), 0]$$

Therefore, Equations (7)–(9) are rewritten as

$$\frac{1}{r}\frac{\partial}{\partial r}\left(r\frac{\partial A_z^{sl,j}}{\partial r}\right) + \frac{1}{r^2}\frac{\partial^2 A_z^{sl,j}}{\partial \theta^2} = -\mu_0 J_z^{sl,j} \tag{10}$$

$$\frac{1}{r}\frac{\partial}{\partial r}\left(r\frac{\partial A_z^{m,k}}{\partial r}\right) + \frac{1}{r^2}\frac{\partial^2 A_z^{m,k}}{\partial \theta^2} = -\frac{\mu_0}{r}\left(M_\theta^k - \frac{\partial M_r^k}{\partial \theta}\right) \tag{11}$$

$$\frac{1}{r}\frac{\partial}{\partial r}\left(r\frac{\partial A_z^{\chi}}{\partial r}\right) + \frac{1}{r^2}\frac{\partial^2 A_z^{\chi}}{\partial \theta^2} = 0, \; \chi = \{(a), (so, j)\} \tag{12}$$

*2.4. Boundary Conditions*

The perpendicular magnetic flux density in two adjacent sub-domains must be equal as mathematically represented as follows:

$$\mathbf{n}.(\mathbf{B}^i - \mathbf{B}^{i+})\mathbf{=}0 \tag{13}$$

In this equation, $\mathbf{B}^i$ is the magnetic flux density in sub-domain $i$, and $\mathbf{B}^{i+}$ is the magnetic flux density in the sub-domain $i+$.

Also, if there is no current between the two adjacent sub-domains, the tangential components of the magnetic field intensity at the boundary of the two sub-domains are equal; this expression is shown mathematically by Relation (14).

$$\mathbf{n} \times (\mathbf{H}^i - \mathbf{H}^{i+}) = 0 \tag{14}$$

In this equation $\mathbf{H}^i$ is the magnetic field intensity of the sub-domain $i$ and $\mathbf{H}^{i+}$ is the magnetic field intensity of sub-domain $i+$.

In both (13) and (14), $\mathbf{n}$ is the perpendicular unit vector to the interface between two adjacent sub-domains.

According to Figure 1, all boundary/interface conditions between sub-domains have been shown from (15) to (25) where $\alpha_r$, $\beta$, and $\delta$ are respectively the magnet arc per pole pitch ratio, the span angle of slot-openings, and the span angle of slots as shown in Figure 1.

| Domain (i) | Domain (i+) | Equation | Border of the Interface | Limit | |
|---|---|---|---|---|---|
| Magnet | Rotor yoke | $H_\theta^{m,k}(r,\theta) = 0$ | $r = R_r$ | $\lvert \theta - \alpha - k\pi/p \rvert \le \alpha_r \pi/2p$ | (15) |
| Magnet | Iron next to the PM pole | $H_r^{m,k}(r,\theta) = 0$ | $\theta = \alpha + k\pi/p \pm \alpha_r \pi/2p$ | $R_r \le r \le R_m$ | (16) |
| Air-gap | Magnet | $B_r^a(r,\theta) = B_r^{m,k}(r,\theta)$ | $r = R_m$ | $\lvert \theta - \alpha - k\pi/p \rvert \le \alpha_r \pi/2p$ | (17) |
| Air-gap | Magnet | $H_\theta^a(r,\theta) = \begin{cases} \sum_{k=0}^{p-1} H_\theta^{m,k}(r,\theta) \\ 0 \end{cases}$ | $r = R_m$ | $\begin{cases} \lvert \theta - \alpha - k\pi/p \rvert \le \alpha_r \pi/2p \\ \text{elsewhere} \end{cases}$ | (18) |
| Slot-opening | Edges of slot-opening | $H_r^{so,j}(r,\theta) = 0$ | $\theta = \theta_j \pm \beta/2$ | $R_s \le r \le R_{so}$ | (19) |
| Air-gap | Slot-opening | $B_r^a(r,\theta) = B_r^{so,j}(r,\theta)$ | $r = R_s$ | $\lvert \theta - \theta_j \rvert \le \frac{\beta}{2}$ | (20) |
| Air-gap | Slot-opening | $H_\theta^a(r,\theta) = \begin{cases} \sum_{j=1}^{Q} H_\theta^{so,j}(r,\theta) \\ 0 \end{cases}$ | $r = R_s$ | $\begin{cases} \lvert \theta - \theta_j \rvert \le \frac{\beta}{2} \\ \text{elsewhere} \end{cases}$ | (21) |
| Slot | Slot-opening | $B_r^{sl,j}(r,\theta) = B_r^{so,j}(r,\theta)$ | $r = R_{so}$ | $\lvert \theta - \theta_j \rvert \le \frac{\beta}{2}$ | (22) |
| Slot | Edge of tooth, Slot-opening, Edge of tooth | $H_\theta^{sl,j}(r,\theta) = \begin{cases} 0 \\ H_\theta^{so,j}(r,\theta) \\ 0 \end{cases}$ | $r = R_{so}$ | $\begin{cases} \theta_j - \frac{\delta}{2} \le \theta < \theta_j - \frac{\beta}{2} \\ \theta_j - \frac{\beta}{2} \le \theta \le \theta_j + \frac{\beta}{2} \\ \theta_j + \frac{\beta}{2} < \theta \le \theta_j + \frac{\delta}{2} \end{cases}$ | (23) |
| Slot | Edges of the slot | $H_r^{sl,j}(r,\theta) = 0$ | $\theta = \theta_j \pm \delta/2$ | $R_{so} \le r \le R_{sl}$ | (24) |
| Slot | Stator yoke | $H_\theta^{sl,j}(r,\theta) = 0$ | $r = R_{sl}$ | $\lvert \theta - \theta_j \rvert \le \frac{\delta}{2}$ | (25) |

## 2.5. Extracting the Fourier Series of the Armature Reaction

To consider the effect of the armature reaction, the current density of each slot should be represented as Fourier series. The current of each phase varies with time and can be represented as Equation (26).

$$I_k(t) = \sum_v I_v \sin[v(p\omega t - \gamma_k) + \theta_v], \ k = 1, 2, \cdots, q \tag{26}$$

where $q$ is the number of the phases, $v$ shows the order of the harmonics, $\omega$ is the angular velocity of the rotor, $p$ is the number of the pole-pairs, $\gamma_k = 2\pi(k-1)/q$ is the time offset of the phase k*th* respect to the first phase. Also $I_v$ and $\theta_v$ are the magnitude and phase offset of *vth* harmonic, respectively.

It is obvious that the relation between the current density in each slot and phase current is dependent on the winding configuration. If the winding configuration is like Figure 2a,b, each slot is considered as one sub-region, like Figure 3a,b. But, if the configuration is similar to Figure 2c, each slot consists of two sub-regions (upper and lower sub-regions), as represented in Figure 3c.

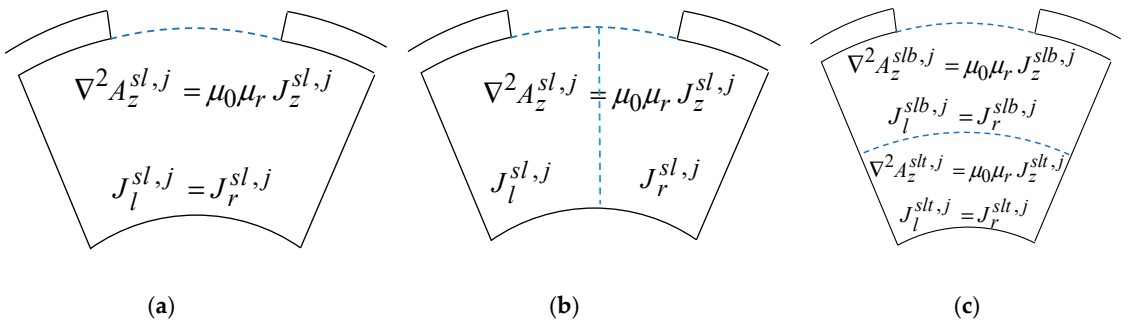

(**a**)  (**b**)  (**c**)

**Figure 3.** Sub-region division according to the winding configuration: (**a**) whole slot is considered as one region and belongs to one coil, (**b**) whole slot is considered as one region and left side and right side of the slot belongs to different coils, (**c**) whole slot divided into upper and lower sub-regions and each part belongs to one coil.

The current density in each sub-region of a slot can be represented in Fourier series form as in Equation (27).

$$J_z{}^j(\theta,t) \quad = J_0^j(t) + \sum_{v=1}^{\infty} J_v^j(t)\cos\left(\tfrac{\pi v}{\delta}(\theta - \theta_j + \delta/2)\right)$$
$$\theta_j - \delta/2 \le \theta \le \theta_j + \delta/2. \tag{27}$$

where $J_0^j(t)$ and $J_v^j(t)$ are as follows:

$$J_0^j(t) = \frac{J_\ell^j(t) + J_r^j(t)}{2} \tag{28}$$

$$J_v^j(t) = \frac{J_\ell^j(t) - J_r^j(t)}{\pi v/2}\sin(\pi v/2) \tag{29}$$

In order to complete the Fourier series of each sub-region of a slot, it is necessary to obtain the current density of phases in each sub-region of a slot at a specific time by Equations (30) and (31):

$$J = \frac{I}{K_f A_{slot}} \tag{30}$$

$$J = \frac{I}{K_f A_{slot}/2} \tag{31}$$

For instance, the current density in each sub-domain of a slot could be as Figure 4a,b.

If the winding configuration is as shown in Figure 2a,c, the figure of the current density in each sub-region of slot, at a time instant will be as Figure 4a, and if the winding configuration is as shown in Figure 2b, the figure of the current density in a time instant will be as Figure 4b. Also if the configuration of the winding is similar to Figure 2c, the current density in each sub-region of the slot will be as Figure 4a.

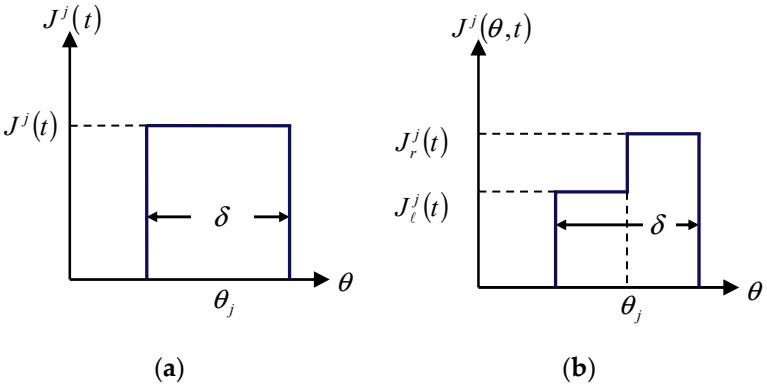

(**a**)　　　　　(**b**)

**Figure 4.** Value of the current density in each sub-region of slot, according to represented winding configurations.

### 2.6. Extracting the Fourier Series of the Magnetization

In the 2-D polar coordinate system, the magnetization vector only has the radial and tangential components as Equation (32).

$$\mathbf{M} = M_r \mathbf{r} + M_\theta \mathbf{\theta} \tag{32}$$

where **r** and $\mathbf{\theta}$ are the radial and tangential unit vectors. $M_r$ and $M_\theta$ are the components of magnetization vector which can be represented as Fourier series expansion of Equations (33) and (34).

$$M_r(\theta_r) = \sum_{n=1,3,5,\dots}^{\infty} M_{rn}\cos(np\theta_r) \tag{33}$$

$$M_\theta(\theta_r) = \sum_{n=1,3,5,\ldots}^{\infty} M_{\theta n} \sin(n p \theta_r) \tag{34}$$

where $M_{rn}$ and $M_{\theta n}$ respectively are the radial and tangential coefficient of the Fourier series and will be determined according to the magnetization pattern (Table 2). In this paper, only the radial magnetization pattern has been used and represented in Figure 5.

**Table 2.** Radial magnetization pattern and its Fourier series components.

| Magnetization Pattern | Illustration | Radial Waveform Component | Tangential Waveform Component | Coefficient of the Radial Component $M_{rw}^{k} = (-1)^{\frac{w-1}{2}+k}\frac{B_{rem}}{\mu_0} \times$ | Coefficient of the Tangential Component $M_{\theta w}^{k} = (-1)^{\frac{w-1}{2}+k}\frac{B_{rem}}{\mu_0} \times$ |
|---|---|---|---|---|---|
| Radial Magnetization | | | | $\frac{4}{w\pi}\sin\left(\frac{w\pi\alpha_p}{2\alpha_r}\right)$ | 0 |

## 2.7. Finding the General Solution

The overall format for the general solution in all sub-domains can be represented as Equation (35):

$$A(r,\theta) = \sum_{n'=1}^{\infty}\left(A_{n'}r^{n'} + B_{n'}r^{n'}\right).\left(C_{n'}\cos n'\theta + D_{n'}\sin n'\theta\right) + (A_0\ln r + B_0)(C_0\theta + D_0) \tag{35}$$

The general solution not only has the capability to satisfy the related PDE, but must satisfy the boundary conditions of the related sub-domain, especially Equations (16), (19), and (24). So the general solutions for sub-domains are as Equations (36)–(40).

The general solution of Poisson equation in slot sub-domain will be as follows:

$$A_z^{sl,j}(r,\theta) = b_0^{sl,j}\ln r + A_p^{sl,j}(r,\theta) + \sum_{v=1}^{\infty}\left[a_v^{sl,j}\left(\frac{r}{R_{so}}\right)^{\frac{\pi v}{\delta}} + b_v^{sl,j}\left(\frac{R_{sl}}{r}\right)^{\frac{\pi v}{\delta}}\right] \\ \times \cos\left(\frac{\pi v}{\delta}\left(\theta - \theta_j + \frac{\delta}{2}\right)\right) \tag{36}$$

The particular solution is as follows:

$$A_p^{sl,j}(r,\theta) = -\frac{\mu_0}{4}J_0^j r^2 + \sum_{v=1}^{\infty}\frac{\mu_0 J_v^j r^2}{\left(\frac{\pi v}{\delta}\right)^2 - 4}\cos\left(\frac{\pi v}{\delta}\left(\theta - \theta_j + \frac{\delta}{2}\right)\right) \tag{37}$$

Also the general solution for the slot-opening sub-domain is

$$A_z^{so,j}(r,\theta) = b_0^{so,j}\ln r + \sum_{u=1}^{\infty}\left[a_u^{so,j}\left(\frac{r}{R_s}\right)^{\frac{\pi u}{\beta}} + b_u^{so,j}\left(\frac{R_{so}}{r}\right)^{\frac{\pi u}{\beta}}\right] \\ \times \cos\left(\frac{\pi u}{\beta}\left(\theta - \theta_j + \frac{\beta}{2}\right)\right) \tag{38}$$

The general solution for the air-gap sub-domain is

$$A_z^{so,j}(r,\theta) = b_0^{so,j}\ln r + \sum_{u=1}^{\infty}\left[a_u^{so,j}\left(\frac{r}{R_s}\right)^{\frac{\pi u}{\beta}} + b_u^{so,j}\left(\frac{R_{so}}{r}\right)^{\frac{\pi u}{\beta}}\right] \\ \times \cos\left(\frac{\pi u}{\beta}\left(\theta - \theta_j + \frac{\beta}{2}\right)\right) \tag{39}$$

And finally, Poisson equation in the PMs sub-domain has the general solution as Equation (40).

$$
\begin{aligned}
A_z^{m,k}(r,\theta) \quad &= b_0^{m,k} \ln r + \sum_{w=1}^{\infty} \left[ a_w^{m,k} \left( \frac{r}{R_r} \right)^{\frac{wp}{\alpha_r}} + b_w^{m,k} \left( \frac{R_m}{r} \right)^{\frac{wp}{\alpha_r}} + k_w^k r \right] \\
&\times \cos \left( \frac{wp}{\alpha_r} \left( \theta - \alpha - \frac{k\pi}{p} + \frac{\alpha_r \pi}{2p} \right) \right)
\end{aligned}
\tag{40}
$$

where

$$
k_w^k = -\mu_0 \chi_w
\begin{cases}
\frac{\frac{wp}{\alpha_r} M_{rw}^k - M_{\theta w}^k}{\left( \frac{wp}{\alpha_r} \right)^2 - 1} & wp \neq \alpha_r \\
\frac{M_{rw}^k - M_{\theta w}^k}{2} \ln r & wp = \alpha_r
\end{cases}
\tag{41}
$$

$$
\chi_w = \frac{1 - (-1)^w}{2}
\tag{42}
$$

In order to simplify the general solution in PM sub-domain, boundary condition (15) has been implemented.

$$
\begin{aligned}
A_z^{m,k}(r,\theta) \quad &= \sum_{w=1}^{\infty} \left\{ b_w^{m,k} \left[ \left( \frac{R_m}{R_r} \right)^{\frac{wp}{\alpha_r}} \left( \frac{r}{R_r} \right)^{\frac{wp}{\alpha_r}} + \left( \frac{R_m}{r} \right)^{\frac{wp}{\alpha_r}} \right] \right. \\
&\left. + R_r \zeta_{w1}^k \left( \frac{r}{R_r} \right)^{\frac{wp}{\alpha_r}} + k_w^k r \right\} \times \cos \left( \frac{wp}{\alpha_r} \left( \theta - \alpha - \frac{k\pi}{p} + \frac{\alpha_r \pi}{2p} \right) \right)
\end{aligned}
\tag{43}
$$

where

$$
\begin{aligned}
\zeta_{w1}^k &= \frac{\alpha_r}{pw} \left( \frac{dk_w^k r}{dr} \Big|_{r=R_r} + \mu_0 \chi_w M_{\theta w}^k \right) = \\
&-\mu_0 \chi_w
\begin{cases}
\frac{M_{rw}^k - \frac{wp}{\alpha_r} M_{\theta w}^k}{\left( \frac{wp}{\alpha_r} \right)^2 - 1} & wp \neq \alpha_r \\
\frac{M_{rw}^k - M_{\theta w}^k}{2} (1 + \ln R_r) - M_{\theta w}^k & wp = \alpha_r .
\end{cases}
\end{aligned}
\tag{44}
$$

Also, by implementing boundary condition (25), the general solution in slot sub-domain will be simplified as in Equation (45).

$$
\begin{aligned}
A_z^{sl,j}(r,\theta) \quad &= \frac{\mu_0}{4} J_0^{sl,j} \left( 2R_{sl}^2 \ln(r) - r^2 \right) + \sum_{v=1}^{V} \left\{ b_v^{sl,j} \left[ \left( \frac{r}{R_{so}} \right)^{\frac{\pi v}{\delta}} + \left( \frac{R_{sl}}{r} \right)^{\frac{\pi v}{\delta}} \left( \frac{R_{sl}}{R_{so}} \right)^{\frac{\pi v}{\delta}} \right] \right. \\
&\left. + \frac{\mu_0 J_v^j}{\left( \frac{\pi v}{\delta} \right)^2 - 4} \left[ r^2 - \frac{2R_{sl}}{\frac{\pi v}{\delta}} \left( \frac{R_{sl}}{r} \right)^{\frac{\pi v}{\delta}} \right] \right\} \times \cos \left( \frac{\pi v}{\delta} \left( \theta - \theta_j + \frac{\delta}{2} \right) \right)
\end{aligned}
\tag{45}
$$

*2.8. Obtaining Integral Coefficients*

For implementing boundary condition (17), the correlation technique must be used [31].

By multiplying Equation (17) in $\frac{2p}{\alpha_r \pi} \sin \left( \frac{wp}{\alpha_r} \left( \theta - \alpha - \frac{k\pi}{p} + \frac{\alpha_r \pi}{2p} \right) \right)$ and integration over $\left[ \alpha + \frac{k\pi}{p} - \frac{\pi \alpha_r}{2p}, \alpha + \frac{k\pi}{p} + \frac{\pi \alpha_r}{2p} \right]$, Equation (46) will be obtained.

$$
\begin{aligned}
&\frac{2p}{\alpha_r \pi} \int_{\alpha + k\pi/p - \alpha_r \pi/2p}^{\alpha + k\pi/p + \alpha_r \pi/2p} B_r^a \Big|_{r=R_m} \sin \left( \frac{wp}{\alpha_r} \left( \theta - \alpha - \frac{k\pi}{p} + \frac{\alpha_r \pi}{2p} \right) \right) d\theta \\
&= \frac{2p}{\alpha_r \pi} \int_{\alpha + k\pi/p - \alpha_r \pi/2p}^{\alpha + k\pi/p + \alpha_r \pi/2p} B_r^{m,k} \Big|_{r=R_m} \sin \left( \frac{wp}{\alpha_r} \left( \theta - \alpha - \frac{k\pi}{p} + \frac{\alpha_r \pi}{2p} \right) \right) d\theta
\end{aligned}
\tag{46}
$$

From Equation (46), Equation (47) will be deduced.

$$
\begin{aligned}
&\frac{wp}{\alpha_r} \left[ 1 + \left( \frac{R_m}{R_r} \right)^{\frac{2wp}{\alpha_r}} \right] a_w^{m,k} - \sum_{n=1}^{N} n \left\{ \left[ a_n^a \left( \frac{R_s}{R_m} \right)^n + b_n^a \right] \sigma_s(n, w, k) \right. \\
&\left. - \left[ c_n^a \left( \frac{R_s}{R_m} \right)^n + d_n^a \right] \sigma_c(n, w, k) \right\} = -R_m \frac{wp}{\alpha_r} \left[ \zeta_{w1}^k \left( \frac{R_m}{R_r} \right)^{\frac{wp}{\alpha_r} + 1} + \zeta_{w2}^k \right]
\end{aligned}
\tag{47}
$$

where

$$
\zeta_{w2}^k = k_w^k \Big|_{r=R_m} = -\mu_0 \chi_w \begin{cases} \dfrac{\frac{wp}{\alpha_r} M_{r\,w}^k - M_{\theta\,w}^k}{\left(\frac{wp}{\alpha_r}\right)^2 - 1} & wp \neq \alpha_r \\ \dfrac{M_{r\,w}^k - M_{\theta\,w}^k}{2} \ln R_m & wp = \alpha_r \end{cases} \tag{48}
$$

$$
\sigma_c(n, w, k) = \frac{2p}{\alpha_r \pi} \int_{\alpha+k\pi/p-\alpha_r\pi/2p}^{\alpha+k\pi/p+\alpha_r\pi/2p} \cos(n\theta) \sin\left(\frac{wp}{\alpha_r}\left(\theta - \alpha - \frac{k\pi}{p} + \frac{\alpha_r\pi}{2p}\right)\right) d\theta \tag{49}
$$

$$
\sigma_s(n, w, k) = \frac{2p}{\alpha_r \pi} \int_{\alpha+k\pi/p-\alpha_r\pi/2p}^{\alpha+k\pi/p+\alpha_r\pi/2p} \sin(n\theta) \sin\left(\frac{wp}{\alpha_r}\left(\theta - \alpha - \frac{k\pi}{p} + \frac{\alpha_r\pi}{2p}\right)\right) d\theta \tag{50}
$$

The solutions of the integrals have been given in the Appendix A.

By using correlation technique, Relation (18) must be multiplied by $\frac{1}{\pi}\sin(n\theta)$ and integration on the interval $[\alpha - \pi, \alpha + \pi]$, Equation (51) will be obtained.

$$
\frac{1}{\pi} \int_{\alpha-\pi}^{\alpha+\pi} H_\theta^a|_{r=R_m} \sin(n\theta)\, d\theta = \frac{1}{\pi} \sum_{k=0}^{2p-1} \int_{\alpha+k\pi/p-\alpha_r\pi/2p}^{\alpha+k\pi/p+\alpha_r\pi/2p} H_\theta^{m,k}\Big|_{r=R_m} \sin(n\theta)\, d\theta \tag{51}
$$

Equation (51) results in Equation (52):

$$
\sum_{k=0}^{2p-1} \sum_{w=1}^{W} \frac{wp}{\mu_r \alpha_r}\left[\left(\frac{R_m}{R_r}\right)^{\frac{2wp}{\alpha_r}} - 1\right] \rho_s(n, w, k)\, a_w^{m,k} + n\left[c_n^a\left(\frac{R_s}{R_m}\right)^n - d_n^a\right]
$$
$$
= \sum_{k=0}^{2p-1} \sum_{w=1}^{W} \frac{wp R_m}{\mu_r \alpha_r}\left[-\zeta_{w1}^k\left(\frac{R_m}{R_r}\right)^{\frac{wp}{\alpha_r}-1} + \zeta_{w3}^k\right]\rho_s(n, w, k) \tag{52}
$$

where

$$
\zeta_{w3}^k = \frac{\alpha_r}{wp}\left(\frac{dk_w^k r}{dr}\Big|_{r=R_m} + \mu_0 \chi_w M_{\theta\,w}^k\right) =
$$
$$
-\mu_0 \chi_w \begin{cases} \dfrac{M_{r\,w}^k - \frac{wp}{\alpha_r} M_{\theta\,w}^k}{\left(\frac{wp}{\alpha_r}\right)^2 - 1} & wp \neq \alpha_r \\ \dfrac{M_{r\,w}^k - M_{\theta\,w}^k}{2}(1 + \ln R_m) - M_{\theta\,w}^k & wp = \alpha_r \end{cases} \tag{53}
$$

$$
\rho_s(n, w, k) = \frac{1}{\pi} \int_{\alpha+k\pi/p-\alpha_r\pi/2p}^{\alpha+k\pi/p+\alpha_r\pi/2p} \cos\left(\frac{wp}{\alpha_r}\left(\theta - \alpha - \frac{k\pi}{p} + \frac{\alpha_r\pi}{2p}\right)\right) \sin(n\theta)\, d\theta \tag{54}
$$

The solution of the integral has been given in the Appendix A.

Again Equation (18) must be multiplied by $\frac{1}{\pi}\cos(n\theta)$, then integration on the interval $[\alpha - \pi, \alpha + \pi]$ causes:

$$
\frac{1}{\pi} \int_{\alpha-\pi}^{\alpha+\pi} H_\theta^a|_{r=R_m} \cos(n\theta)\, d\theta = \frac{1}{\pi} \sum_{k=0}^{2p-1} \int_{\alpha+k\pi/p-\alpha_r\pi/2p}^{\alpha+k\pi/p+\alpha_r\pi/2p} H_\theta^{m,k}\Big|_{r=R_m} \cos(n\theta)\, d\theta \tag{55}
$$

Equation (55) results in Equation (56).

$$
\sum_{k=0}^{2p-1} \sum_{w=1}^{W} \frac{wp}{\mu_r \alpha_r}\left[\left(\frac{R_m}{R_r}\right)^{\frac{2wp}{\alpha_r}} - 1\right] \rho_c(n, w, k)\, a_w^{m,k} + n\left[a_n^a\left(\frac{R_s}{R_m}\right)^n - b_n^a\right]
$$
$$
= \sum_{k=0}^{2p-1} \sum_{w=1}^{W} \frac{wp R_m}{\mu_r \alpha_r}\left[-\zeta_{w1}^k\left(\frac{R_m}{R_r}\right)^{\frac{wp}{\alpha_r}-1} + \zeta_{w3}^k\right]\rho_c(n, w, k) \tag{56}
$$

where

$$
\rho_c(n, w, k) = \frac{1}{\pi} \int_{\alpha+k\pi/p-\alpha_r\pi/2p}^{\alpha+k\pi/p+\alpha_r\pi/2p} \cos\left(\frac{wp}{\alpha_r}\left(\theta - \alpha - \frac{k\pi}{p} + \frac{\alpha_r\pi}{2p}\right)\right) \cos(n\theta)\, d\theta \tag{57}
$$

The solution of the integral has been given in the Appendix A.

For implementing boundary condition (20), multiplying $\frac{2}{\beta} \sin\left(\frac{\pi u}{\beta}\left(\theta - \theta_j + \frac{\beta}{2}\right)\right)$ to Equation (20) and integration over $\left[\theta_j - \frac{\beta}{2}, \theta_j + \frac{\beta}{2}\right]$ yields

$$
\begin{aligned}
&\frac{2}{\beta} \int_{\theta_j - \beta/2}^{\theta_j + \beta/2} B_r^{so,j}(r,\theta)\Big|_{r=R_s} \sin\left(\frac{\pi u}{\beta}\left(\theta - \theta_j + \frac{\beta}{2}\right)\right) d\theta = \\
&\frac{2}{\beta} \int_{\theta_j - \beta/2}^{\theta_j + \beta/2} B_r^a(r,\theta)\Big|_{r=R_s} \sin\left(\frac{\pi u}{\beta}\left(\theta - \theta_j + \frac{\beta}{2}\right)\right) d\theta
\end{aligned}
\tag{58}
$$

Equation (59) is obtained via the simplification of Equation (58).

$$
\begin{aligned}
&-\sum_{n=1}^{N} n\left\{ \left[a_n^a + b_n^a\left(\frac{R_s}{R_m}\right)^n\right]\varepsilon_s(n,u,j) - \left[c_n^a + d_n^a\left(\frac{R_s}{R_m}\right)^n\right]\varepsilon_c(n,u,j)\right\} \\
&+\frac{\pi u}{\beta}\left(\frac{R_{so}}{R_s}\right)^{\frac{\pi u}{\beta}} a_u^{so,j} + \frac{\pi u}{\beta} b_u^{so,j} = 0
\end{aligned}
\tag{59}
$$

where

$$
\varepsilon_s(n,u,j) = \frac{2}{\beta} \int_{\theta_j - \beta/2}^{\theta_j + \beta/2} \sin(n\theta) \sin\left(\frac{\pi u}{\beta}\left(\theta - \theta_j + \frac{\beta}{2}\right)\right) d\theta
\tag{60}
$$

$$
\varepsilon_c(n,u,j) = \frac{2}{\beta} \int_{\theta_j - \beta/2}^{\theta_j + \beta/2} \cos(n\theta) \sin\left(\frac{\pi u}{\beta}\left(\theta - \theta_j + \frac{\beta}{2}\right)\right) d\theta
\tag{61}
$$

The solution of the integral has been given in the Appendix A.

The correlation technique is used for boundary condition (21) and $\frac{1}{\pi}\cos(n\theta)$ is multiplied to Equation (21) and integration is taken over $[-\pi, \pi]$.

$$
\frac{1}{\pi} \int_{-\pi}^{\pi} H_\theta^a(r,\theta)|_{r=R_s} \cos(n\theta)\, d\theta = \frac{1}{\pi}\sum_{j=1}^{Q} \int_{\theta_j - \beta/2}^{\theta_j + \beta/2} H_\theta^{so,j}(r,\theta)\Big|_{r=R_s} \cos(n\theta)\, d\theta
\tag{62}
$$

Simplifying Equation (62) yields (63).

$$
\begin{aligned}
&n\left[a_n^a - b_n^a\left(\frac{R_s}{R_m}\right)^n\right] - \sum_{j=1}^{Q}\sum_{u=1}^{U} \frac{\pi u}{\beta}\left[\left(\frac{R_{so}}{R_s}\right)^{\frac{\pi u}{\beta}} a_u^{so,j} - b_u^{so,j}\right]\eta_c(n,u,j) \\
&-\sum_{j=1}^{Q} \eta_c(n,0,j) b_0^{so,j} = 0
\end{aligned}
\tag{63}
$$

where

$$
\eta_c(n,u,j) = \frac{1}{\pi} \int_{\theta_j - \beta/2}^{\theta_j + \beta/2} \cos(n\theta) \cos\left(\frac{\pi u}{\beta}\left(\theta - \theta_j + \frac{\beta}{2}\right)\right) d\theta
\tag{64}
$$

Also, by multiplying Equation (21) to $\frac{1}{\pi}\sin(n\theta)$ and integration over interval $[-\pi, \pi]$ the following expression is obtained:

$$
\frac{1}{\pi} \int_{-\pi}^{\pi} H_\theta^a(r,\theta)|_{r=R_s} \sin(n\theta)\, d\theta = \frac{1}{\pi}\sum_{j=1}^{Q} \int_{\theta_j - \beta/2}^{\theta_j + \beta/2} H_\theta^{so,j}(r,\theta)\Big|_{r=R_s} \sin(n\theta)\, d\theta
\tag{65}
$$

Simplification of Equation (65) causes the formation of Equation (66).

$$n\left[c_n^a - \left(\frac{R_s}{R_m}\right)^n d_n^a\right] - \sum_{j=1}^{Q}\sum_{u=1}^{U}\frac{\pi u}{\beta}\left[\left(\frac{R_{so}}{R_s}\right)^{\frac{\pi u}{\beta}}a_u^{so,j} - b_u^{so,j}\right]\eta_s(n,u,j)$$
$$- \sum_{j=1}^{Q}\eta_s(n,0,j)\,b_0^{so,j} = 0 \tag{66}$$

where

$$\eta_s(n,u,j) = \frac{1}{\pi}\int\limits_{\theta_j-\beta/2}^{\theta_j+\beta/2}\sin(n\theta)\cos\left(\frac{\pi u}{\beta}\left(\theta - \theta_j + \frac{\beta}{2}\right)\right)d\theta \tag{67}$$

The correlation technique is used and boundary condition (22) is multiplied by $\frac{2}{\beta}\sin\left(\frac{\pi u}{\beta}\left(\theta - \theta_j + \frac{\beta}{2}\right)\right)$. Integration over interval $\left[\theta_j - \frac{\beta}{2},\ \theta_j + \frac{\beta}{2}\right]$ results in Equation (68).

$$\frac{2}{\beta}\int\limits_{\theta_j-\beta/2}^{\theta_j+\beta/2}B_r^{sl,j}(r,\theta)\Big|_{r=R_{so}}\sin\left(\frac{\pi u}{\beta}\left(\theta - \theta_j + \frac{\beta}{2}\right)\right)d\theta =$$
$$\frac{2}{\beta}\int\limits_{\theta_j-\beta/2}^{\theta_j+\beta/2}B_r^{so,j}(r,\theta)\Big|_{r=R_{so}}\sin\left(\frac{\pi u}{\beta}\left(\theta - \theta_j + \frac{\beta}{2}\right)\right)d\theta \tag{68}$$

Simplifying (68) results in (69).

$$\frac{\pi u}{\beta}\left[a_u^{so,j} + \left(\frac{R_{so}}{R_s}\right)^{\frac{\pi u}{\beta}}b_u^{so,j}\right] - \sum_{v=1}^{V}\frac{\pi v}{\delta}\left[\left(\frac{R_{sl}}{R_{so}}\right)^{\frac{2\pi v}{\delta}}+1\right]\gamma_s(u,v)\,b_v^{sl,j}$$
$$= \sum_{v=1}^{V}\frac{\mu_0 J_v^j}{\left(\frac{\pi v}{\delta}\right)^2-4}\left[\frac{\pi v}{\delta}R_{sl}^2 - 2R_{so}^2\left(\frac{R_{sl}}{R_{so}}\right)^{\frac{\pi v}{\delta}}\right]\gamma_s(u,v) \tag{69}$$

where

$$\gamma_s(u,v) = \frac{2}{\beta}\int\limits_{\theta_j-\beta/2}^{\theta_j+\beta/2}\sin\left(\frac{\pi v}{\delta}\left(\theta - \theta_j + \frac{\delta}{2}\right)\right)\sin\left(\frac{\pi u}{\beta}\left(\theta - \theta_j + \frac{\beta}{2}\right)\right)d\theta \tag{70}$$

Solution of the integral are given in the Appendix A.

Correlation technique is implemented for boundary condition (23) and it is multiplied by $\frac{2}{\delta}\cos\left(\frac{\pi v}{\delta}\left(\theta - \theta_j + \frac{\delta}{2}\right)\right)$. Integration over interval $\left[\theta_j - \frac{\delta}{2},\ \theta_j + \frac{\delta}{2}\right]$ yields Equation (71).

$$\frac{2}{\delta}\int\limits_{\theta_j-\delta/2}^{\theta_j+\delta/2}H_\theta^{sl,j}(r,\theta)\Big|_{r=R_{so}}\cos\left(\frac{\pi v}{\delta}\left(\theta - \theta_j + \frac{\delta}{2}\right)\right)d\theta =$$
$$\frac{2}{\delta}\int\limits_{\theta_j-\beta/2}^{\theta_j+\beta/2}H_\theta^{so,j}(r,\theta)\Big|_{r=R_{so}}\cos\left(\frac{\pi v}{\delta}\left(\theta - \theta_j + \frac{\delta}{2}\right)\right)d\theta \tag{71}$$

Simplifying Equation (71) results in Equation (72).

$$\sum_{u=1}^{U}\frac{\pi u}{\beta}\left[-a_u^{so,j} + \left(\frac{R_{so}}{R_s}\right)^{\frac{\pi u}{\beta}}b_u^{so,j}\right]\gamma_c(u,v) - \gamma_c(0,v)\,b_0^{so,j} + \frac{\pi v}{\delta}\left[\left(\frac{R_{sl}}{R_{so}}\right)^{\frac{2\pi v}{\delta}}-1\right]b_v^{sl,j}$$
$$= \frac{-2\mu_0 J_v^j}{\left(\frac{\pi v}{\delta}\right)^2-4}\left[R_{sl}^2 - R_{so}^2\left(\frac{R_{sl}}{R_{so}}\right)^{\frac{\pi v}{\delta}}\right] \tag{72}$$

where

$$\gamma_c(u,v) = \frac{2}{\delta} \int\limits_{\theta_j-\beta/2}^{\theta_j+\beta/2} \cos\left(\frac{\pi v}{\delta}\left(\theta - \theta_j + \frac{\delta}{2}\right)\right) \cos\left(\frac{\pi u}{\beta}\left(\theta - \theta_j + \frac{\beta}{2}\right)\right) d\theta \tag{73}$$

Integration over interval $\left[\theta_j - \frac{\delta}{2}, \theta_j + \frac{\delta}{2}\right]$ on boundary condition (23) causes Equation (74).

$$\int\limits_{\theta_j-\delta/2}^{\theta_j+\delta/2} H_\theta^{sl,j}(r,\theta)\Big|_{r=R_{so}} d\theta = \int\limits_{\theta_j-\beta/2}^{\theta_j+\beta/2} H_\theta^{so,j}(r,\theta)\Big|_{r=R_{so}} d\theta \tag{74}$$

From Equation (74), Equation (75) will be deduced.

$$b_0^{so,j} = \frac{\mu_0 J_0^j}{2}\left(R_{so}^2 - R_{sl}^2\right)\frac{\delta}{\beta} \tag{75}$$

### 2.9. Overlapping Winding

In overlapping winding, each slot is divided into two sub-domains as represented in Figure 3c. The upper and lower sub-domains respectively are indicated with *slb* and *slt* indices. Hence Equation (69) will be as follows:

$$\frac{\pi u}{\beta}\left[a_u^{so,j} + \left(\frac{R_{so}}{R_s}\right)^{\frac{\pi u}{\beta}} b_u^{so,j}\right] - \sum_{v=1}^{V} \frac{\pi v}{\delta}\left[\frac{(R_{sl}/R_{so})^{\frac{2\pi v}{\delta}} + 1}{(R_{slm}/R_{so})^{\frac{\pi v}{\delta}}}\right]\gamma_s(u,v)b_v^{slb,j} = 0 \tag{76}$$

Also, Equations (72) and (75) will be modified as (77) and (78) respectively.

$$\sum_{u=1}^{U} \frac{\pi u}{\beta}\left[-a_u^{so,j} + \left(\frac{R_{so}}{R_s}\right)^{\frac{\pi u}{\beta}} b_u^{so,j}\right]\gamma_c(u,v) + \frac{\pi v}{\delta}\left[\frac{(R_{sl}/R_{so})^{\frac{2\pi v}{\delta}} - 1}{(R_{slm}/R_{so})^{\frac{\pi v}{\delta}}}\right]b_v^{slb,j}$$
$$-\gamma_c(0,v)b_0^{so,j} = 0 \tag{77}$$

$$b_0^{so,j} = \frac{\mu_0}{2}\left[J_{b0}^j\left(R_{slm}^2 - R_{sl}^2\right) + J_{t0}^j\left(R_{so}^2 - R_{slm}^2\right)\right]\frac{\delta}{\beta} \tag{78}$$

where $R_{slm} = \sqrt{(R_{sl}^2 + R_{so}^2)/2}$ is the radii of middle of the slot which divides it into two equal areas. $J_{b0}^j$ and $J_{t0}^j$ are current densities in the lower and upper sub-domains in a slot.

## 3. Quantities

### 3.1. Flux Density

The air-gap flux density vector is one of the most important quantities required for the calculation of other quantities. For obtaining the air-gap flux density, Equation (5) is expanded and Relations (78) and (79) in 2-D polar coordinates are deduced.

$$B_r^a(r,\theta) = \frac{1}{r}\frac{\partial A_z^a}{\partial \theta} = -\sum_{n=1}^{N} n\left\{\left[\frac{a_n^a}{R_m}\left(\frac{r}{R_m}\right)^{n-1} + \frac{b_n^a}{R_s}\left(\frac{R_s}{r}\right)^{n+1}\right]\sin(n\theta)\right.$$
$$\left. -\left[\frac{c_n^a}{R_m}\left(\frac{r}{R_m}\right)^{n-1} + \frac{d_n^a}{R_s}\left(\frac{R_s}{r}\right)^{n+1}\right]\cos(n\theta)\right\} \tag{79}$$

$$B_\theta^a(r,\theta) = -\frac{\partial A_z^a}{\partial r} = -\sum_{n=1}^{N} n\left\{\left[\left[\frac{a_n^a}{R_m}\left(\frac{r}{R_m}\right)^{n-1} - \frac{b_n^a}{R_s}\left(\frac{R_s}{r}\right)^{n+1}\right]\right]\cos(n\theta)\right.$$
$$\left. +\left[\frac{c_n^a}{R_m}\left(\frac{r}{R_m}\right)^{n-1} - \frac{d_n^a}{R_s}\left(\frac{R_s}{r}\right)^{n+1}\right]\sin(n\theta)\right\} \tag{80}$$

### 3.2. Inductances

For calculation of the inductances, just the flux produced by armature is considered. Inductances between phase $k$ and $k'$ are obtained as follows:

$$L_{k,k'} = \sum_{j \in k \ \& \ j' \in k'} \frac{\lambda_{j,j'}}{i_{j'}} \tag{81}$$

where $j = 1, 2, \ldots, Q$ and $j' = 1, 2, \ldots, Q$ are the indices of the coils. $i_{j'}$ is the current of phase $k'$ and $\lambda_{j,j'}$ is the flux linked by coil $j$, which is produced by coil $j'$. If $k = k'$, self-inductance of the phase is calculated.

### 3.3. Back-EMF

In order to calculate the no-load back-EMF of a phase, the permanent magnet flux linked by coils must be calculated by Equation (82).

$$\varphi = \int \mathbf{B} \cdot d\mathbf{S} \tag{82}$$

According to Faraday's law, induced voltage in $j$th coil obtain by Equation (83).

$$E_j = -N_t \omega \frac{d\varphi_j}{d\alpha} \tag{83}$$

where $N_t$ is the number of turns of the coil, $\omega$ and $\alpha$ are the angular velocity and rotor position respectively. The total back-EMF of a phase depends on coils connections.

### 3.4. Instantaneous Electromagnetic Torque

Instantaneous torque consists of cogging torque ($T_{cog}$), electromagnetic torque ($T_{em}$) and reluctance torque ($T_{rel}$).

$$T(t) = T_{cog}(t) + T_{em}(t) + T_{rel}(t) \tag{84}$$

By using Maxwell stress tensor, the instantaneous electromagnetic torque can be obtained as follows:

$$T(t) = \int \int \frac{1}{\mu_0} B_r B_\theta \, ds \tag{85}$$

By expanding Equation (85), Relations (86) and (87) will be obtained.

$$T(t) = L_s \int_{-\pi}^{\pi} \frac{1}{\mu_0} \left( B_{r,PM}^a + B_{r,AR}^a \right) \left( B_{\theta,PM}^a + B_{\theta,AR}^a \right) \big|_{r=R_c} R_c^2 \, d\theta \tag{86}$$

$$T(t) = \frac{L_s R_c^2}{\mu_0} \int_{-\pi}^{\pi} \left( B_{r,PM}^a B_{\theta,PM}^a + B_{r,AR}^a B_{\theta,PM}^a + B_{r,PM}^a B_{\theta,AR}^a + B_{r,AR}^a B_{\theta,AR}^a \right) \big|_{r=R_c} \, d\theta \tag{87}$$

where $B_{r,PM}^a$ and $B_{\theta,PM}^a$ respectively are the radial and tangential flux density components in the air-gap, due to the PMs. Also $B_{r,AR}^a$ and $B_{\theta,AR}^a$ are the radial and tangential magnetic flux density components due to the armature reaction in the air-gap. The parameter $R_c$ is the radius of the middle of inner air-gap.

### 3.5. Unbalanced Magnetic Force

The radial and tangential components of the local traction exerted on the rotor surface can be obtained by Maxwell stress tensor as follows:

$$f_r = \frac{1}{2\mu_0}\left(B_r^2 - B_\theta^2\right) \tag{88}$$

$$f_\theta = \frac{1}{\mu_0}B_r B_\theta \tag{89}$$

By transforming these local tractions to the Cartesian plane, and summation of the same directions, Equations (90)–(93) will be obtained.

$$f_x = f_r \cos\theta - f_\theta \sin\theta \tag{90}$$

$$f_y = f_r \sin\theta + f_\theta \cos\theta \tag{91}$$

$$F_x(t) = \int_{-L/2}^{L/2}\int_{-\pi}^{\pi} f_x\, r\, d\theta\, dz = L\int_{-\pi}^{\pi} f_x\, r\, d\theta \tag{92}$$

$$F_y(t) = \int_{-L/2}^{L/2}\int_{-\pi}^{\pi} f_y\, r\, d\theta\, dz = L\int_{-\pi}^{\pi} f_y\, r\, d\theta \tag{93}$$

Finally, the amplitude of the unbalanced magnetic force can be obtained by Equation (94).

$$F_r = |F(t)| = \sqrt{F_x^2(t) + F_y^2(t)} \tag{94}$$

## 4. Results

In order to investigate the efficacy of the model, a case study with the parameters listed in Table 3 has been used.

Also, the winding configuration for this case study has been shown in Figure 5.

**Table 3.** Parameters used in the case study.

| Parameters | Unit | Symbol | Value |
|---|---|---|---|
| Number of phases | | $q$ | 3 |
| Number of the pole-pair | | $p$ | 4 |
| Number of slots | | $Q$ | 9 |
| Outer radius of the slots | (mm) | $R_{sl}$ | 18 |
| Outer radius of the slot-opening | (mm) | $R_{so}$ | 29 |
| Stator radius | (mm) | $R_s$ | 31 |
| Magnet radius | (mm) | $R_m$ | 32 |
| Radius of the rotor back iron | (mm) | $R_r$ | 38 |
| Axial length | (m) | $L_s$ | 0.1 |
| Span of the slot | (rad) | $\delta$ | 0.6 |
| Span of the slot-opening | (rad) | $\beta$ | 0.3 |
| Pole arc to pole pith of the magnet | | $\alpha_p$ | 0.85 |
| Ratio of the rotor back iron to the pole pitch | | $\alpha_r$ | 0.85 |
| Remanence of magnet | (T) | $B_{rem}$ | 1 |
| Relative permeability of the magnet | | $\mu_r$ | 1.05 |
| Number of harmonics in each sub-domain | | $N, U, V, W$ | 100 |

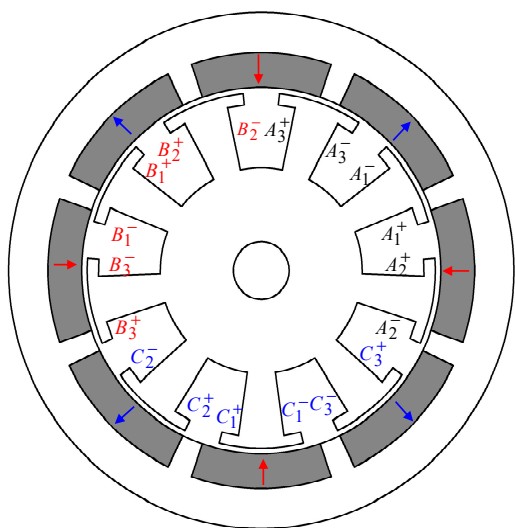

**Figure 5.** The machine topology and winding configuration.

## 4.1. Flux Density

The radial and tangential magnetic flux density components due to the open-circuit and armature reaction are respectively depicted in Figures 6–9. The numerical results obtained from FEM are shown respectively in each figure and confirm the accuracy of the proposed model.

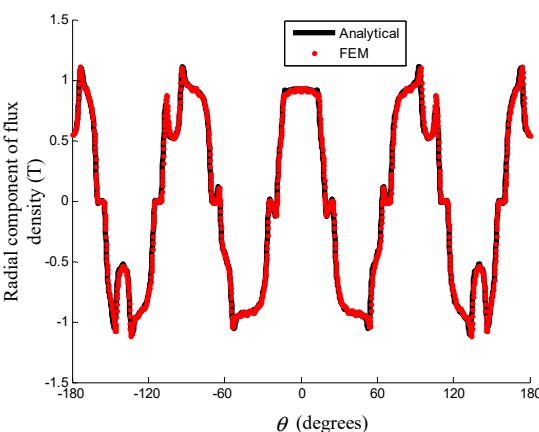

**Figure 6.** Radial magnetic flux density due to just PMs in the middle of the air-gap, when the rotor position is set to zero.

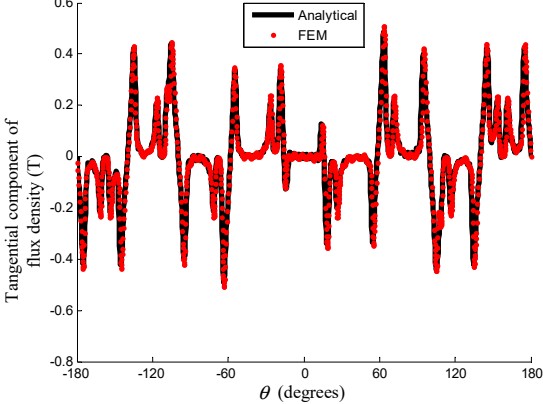

**Figure 7.** Tangential magnetic flux density due to just PMs in the middle of the air-gap, when the rotor position is set to zero.

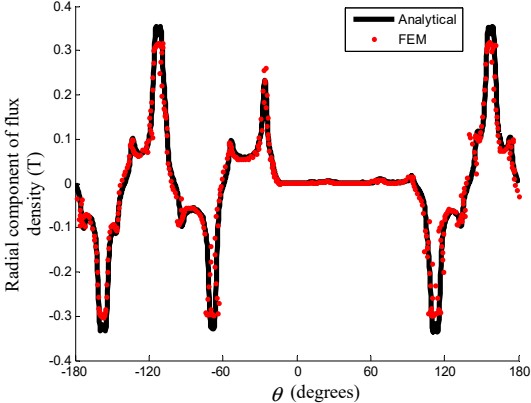

**Figure 8.** Radial magnetic flux density due to just armature winding in the middle of the air-gap, when the current density of phase A is zero, current density of phase B is 4.33 A/mm$^2$ and phase C is 4.33 A/mm$^2$.

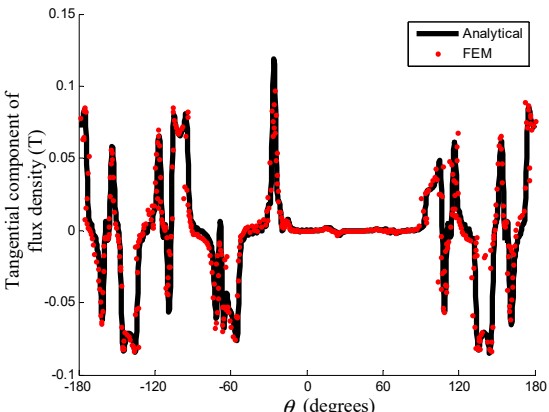

**Figure 9.** Tangential magnetic flux density due to just armature winding in the middle of the air-gap, when the current density of phase A is zero, current density of phase B is 4.33 A/mm$^2$ and phase C is 4.33 A/mm$^2$.

*4.2. Torque*

Instantaneous electromagnetic torque, reluctance torque and cogging torque of the machine have been depicted in Figures 10–12. Both analytic and numeric methods show good agreement which confirms the efficacy of the proposed model.

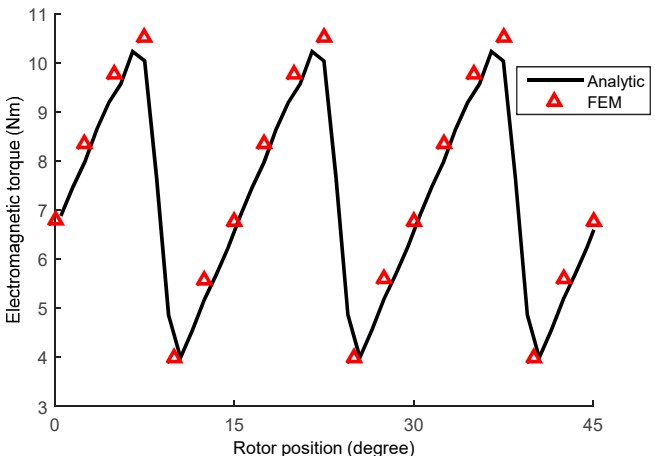

**Figure 10.** Electromagnetic torque vs. rotor position.

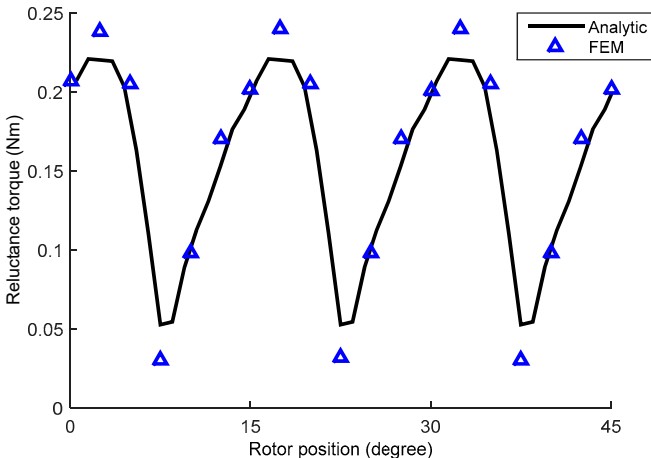

**Figure 11.** Reluctance torque vs. rotor position.

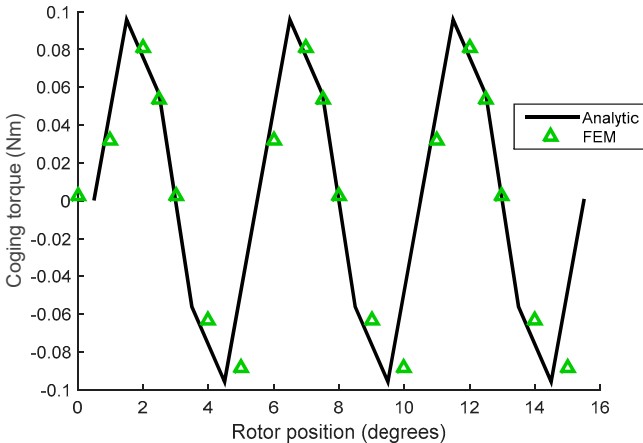

**Figure 12.** Cogging torque vs. rotor position.

### 4.3. Back-EMF and Inductance

Results of the phase back-EMF and line back-EMF are shown in Figure 13. Again it is shown that both analytic and numeric results have good conformity.

Also, the self and mutual inductances have been depicted in Figure 14.

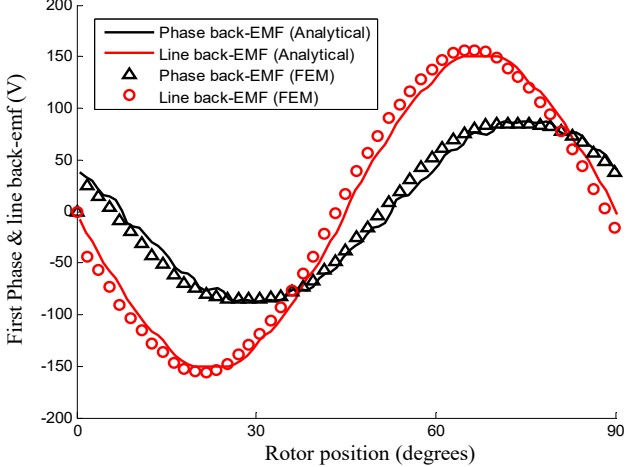

**Figure 13.** Back-electromotive force (EMF) of the first phase and first line.

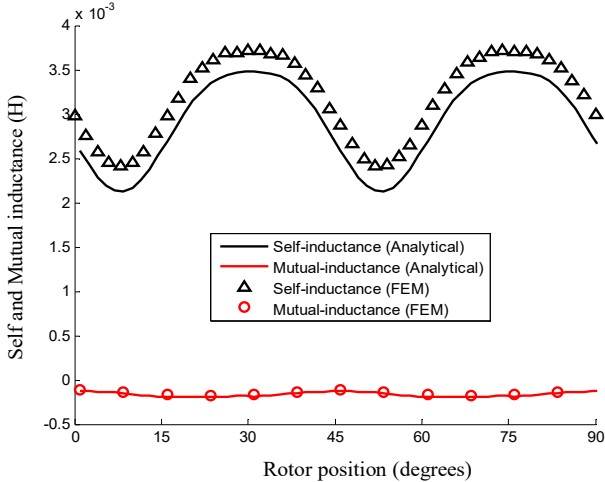

**Figure 14.** Self and mutual inductance of a phase.

## 4.4. Unbalanced Magnetic Force (UMF)

Unbalanced magnetic forces due to the open-circuit, armature reaction, and both of them have been depicted in Figures 15–17. As evident from these figures, unbalanced forces due to the armature reaction exert considerable forces compared to those of the open circuit.

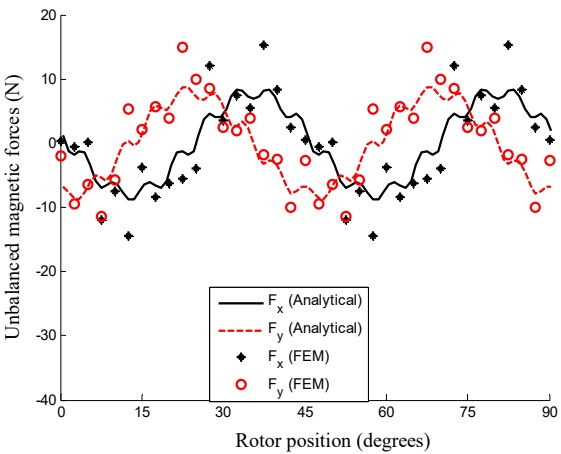

**Figure 15.** Unbalanced magnetic forces just due to the PMs.

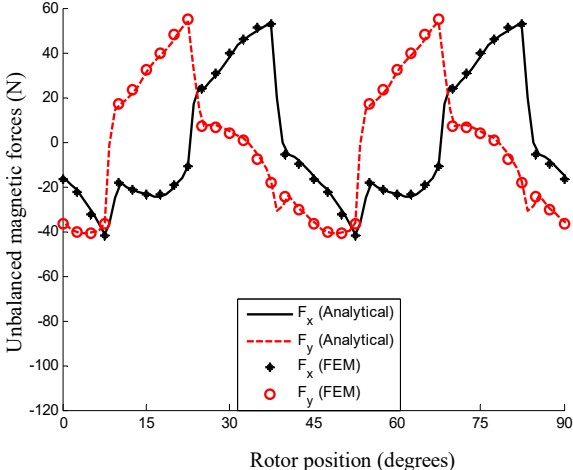

**Figure 16.** Unbalanced magnetic forces just due to the armature reaction.

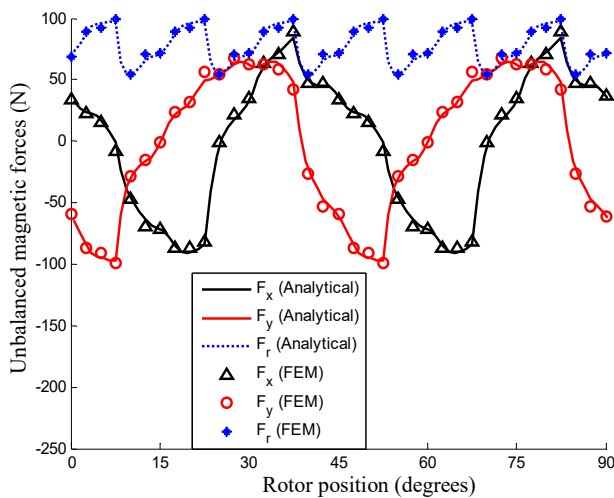

**Figure 17.** Total unbalanced magnetic forces exerted on the rotor.

Both analytic and numeric results are shown in Figures 15–17 and confirm the correctness of the proposed model.

## 5. Conclusions

A 2-D analytical magnetic model is presented for brushless synchronous outer rotor machines with surface inset PMs. For this purpose, Maxwell's equations in the form of the Laplace and Poisson equations are solved in predefined sub-domains of the 2-D polar coordinates. The general and particular solutions for each sub-region are presented so that they have the capability to satisfy the governing PDE and related boundary conditions. Finally, by imposing boundary conditions and solving simultaneous linear algebraic equations, all important quantities such as magnetic flux density, electromagnetic torque, UMF, back-EMF, and inductances are calculated and validated by those obtained by FEM. The results of the analytical model show the efficacy of the proposed approach.

**Author Contributions:** A.V. has done formal analysis, investigation, methodology, writing and editing the original draft. A.R. has done supervision, conceptualization, project administration and reviewing the original draft. H.M.-J. has done formal analysis, investigation, methodology, software, and validation. A.G. has done review and editing.

**Conflicts of Interest:** The authors declare no conflict of interest.

## Appendix A

To define the left and right side current density in each slot for the two layers non-overlapping concentrated winding topology, we have:

$$J_r = \frac{i(t)}{K_f A_w/2} \mathbf{C}_r \tag{A1}$$

$$J_l = \frac{i(t)}{K_f A_w/2} \mathbf{C}_l \tag{A2}$$

where $J_r = [J_r^1 \dots J_r^j \dots J_r^Q]$, $J_l = [J_l^1 \dots J_l^j \dots J_l^Q]$, $i(t) = [i_a\, i_b\, i_c]$, $K_f$ is the filling factor and $A_w$ is the slot area. On the other hand, $\mathbf{C}_r$ and $\mathbf{C}_l$ are as follows:

$$\mathbf{C}_r(i,j) = \begin{cases} \frac{\mathbf{C}(i,j)+|\mathbf{C}(i,j)|}{2} & if \quad -1 < \mathbf{C}(i,j) < 1 \\ 1 & if \quad \mathbf{C}(i,j) = 2 \\ -1 & if \quad \mathbf{C}(i,j) = -2 \end{cases} \tag{A3}$$

$$\mathbf{C}_l(i,j) = \begin{cases} \frac{\mathbf{C}(i,j) - |\mathbf{C}(i,j)|}{2} & if \quad -1 < \mathbf{C}(i,j) < 1 \\ 1 & if \quad \mathbf{C}(i,j) = 2 \\ -1 & if \quad \mathbf{C}(i,j) = -2 \end{cases} \tag{A4}$$

According to the winding topology, **C** is as follows:

$$\mathbf{C} = \begin{bmatrix} 2 & -2 & 1 & 0 & 0 & 0 & 0 & 0 & -1 \\ 0 & 0 & -1 & 2 & -2 & 1 & 0 & 0 & 0 \\ 0 & 0 & 0 & 0 & 0 & -1 & 2 & -2 & 1 \end{bmatrix}_{3 \times Q = 3 \times 9} \tag{A5}$$

where $\mathbf{C}(i,j) = 2$ or $-2$ means slot $j$ accommodates two sides of two coils of phase $i$, which respectively carry positive or negative current. $\mathbf{C}(i,j) = 1$ or $-1$ means slot $j$ accommodates one side of one coil of phase $i$, which respectively carry positive or negative current. Also, 0 means there is no coil of the phase $i$ th in the slot.

For $\alpha_r n \neq wp$:

$$\rho_s(n, w, k) = \frac{\alpha_r}{2\pi} \left\{ \frac{-\cos\left(w\pi + \frac{n\pi\alpha_r}{2p} + n\alpha + \frac{kn\pi}{p}\right) + \cos\left(\frac{n\pi\alpha_r}{2p} - n\alpha - \frac{kn\pi}{p}\right)}{\alpha_r n + wp} \right. $$
$$\left. - \frac{\cos\left(w\pi - \frac{n\pi\alpha_r}{2p} - n\alpha - \frac{kn\pi}{p}\right) - \cos\left(\frac{n\pi\alpha_r}{2p} - n\alpha - \frac{kn\pi}{p}\right)}{\alpha_r n - wp} \right\} \tag{A6}$$

$$\rho_c(n, w, k) = \frac{\alpha_r}{2\pi} \left\{ \frac{\sin\left(w\pi + \frac{n\pi\alpha_r}{2p} + n\alpha + \frac{kn\pi}{p}\right) + \sin\left(\frac{n\pi\alpha_r}{2p} - n\alpha - \frac{kn\pi}{p}\right)}{\alpha_r n + wp} \right. $$
$$\left. - \frac{\sin\left(w\pi - \frac{n\pi\alpha_r}{2p} - n\alpha - \frac{kn\pi}{p}\right) - \sin\left(\frac{n\pi\alpha_r}{2p} - n\alpha - \frac{kn\pi}{p}\right)}{\alpha_r n - wp} \right\} \tag{A7}$$

$$\sigma_s(n, w, k) = \frac{p}{\pi} \left\{ \frac{-\sin\left(w\pi + \frac{n\pi\alpha_r}{2p} + n\alpha + \frac{kn\pi}{p}\right) - \sin\left(\frac{n\pi\alpha_r}{2p} - n\alpha - \frac{kn\pi}{p}\right)}{\alpha_r n + wp} \right. $$
$$\left. - \frac{\sin\left(w\pi - \frac{n\pi\alpha_r}{2p} - n\alpha - \frac{kn\pi}{p}\right) - \sin\left(\frac{n\pi\alpha_r}{2p} - n\alpha - \frac{kn\pi}{p}\right)}{\alpha_r n - wp} \right\} \tag{A8}$$

$$\sigma_c(n, w, k) = \frac{p}{\pi} \left\{ \frac{-\cos\left(w\pi + \frac{n\pi\alpha_r}{2p} + n\alpha + \frac{kn\pi}{p}\right) + \cos\left(\frac{n\pi\alpha_r}{2p} - n\alpha - \frac{kn\pi}{p}\right)}{\alpha_r n + wp} \right. $$
$$\left. + \frac{\cos\left(w\pi - \frac{n\pi\alpha_r}{2p} - n\alpha - \frac{kn\pi}{p}\right) - \cos\left(\frac{n\pi\alpha_r}{2p} - n\alpha - \frac{kn\pi}{p}\right)}{\alpha_r n - wp} \right\} \tag{A9}$$

For $\alpha_r n = wp$:

$$\rho_s(n, w, k) = \frac{-1}{4n\pi} \left[\cos\left(3w\pi/2 + n\alpha + \frac{kn\pi}{p}\right) - \cos\left(w\pi/2 - n\alpha - \frac{kn\pi}{p}\right)\right] $$
$$- \frac{\alpha_r}{2p} \sin\left(w\pi/2 - n\alpha - \frac{kn\pi}{p}\right) \tag{A10}$$

$$\rho_c(n, w, k) = \frac{1}{4n\pi} \left[\sin\left(3w\pi/2 + n\alpha + \frac{kn\pi}{p}\right) + \sin\left(w\pi/2 - n\alpha - \frac{kn\pi}{p}\right)\right] $$
$$+ \frac{\alpha_r}{2p} \cos\left(w\pi/2 - n\alpha - \frac{kn\pi}{p}\right) \tag{A11}$$

$$\sigma_s(n, w, k) = \frac{-1}{2w\pi} \left[\sin\left(3w\pi/2 + n\alpha + \frac{kn\pi}{p}\right) + \sin\left(w\pi/2 - n\alpha - \frac{kn\pi}{p}\right)\right] $$
$$+ \cos\left(w\pi/2 - n\alpha - \frac{kn\pi}{p}\right) \tag{A12}$$

$$\sigma_c(n, w, k) = \frac{-1}{2w\pi} \left[\cos\left(3w\pi/2 + n\alpha + \frac{kn\pi}{p}\right) - \cos\left(w\pi/2 - n\alpha - \frac{kn\pi}{p}\right)\right] $$
$$+ \sin\left(w\pi/2 - n\alpha - \frac{kn\pi}{p}\right) \tag{A13}$$

For $\pi u \neq \beta n$:

$$\varepsilon_s(n,u,j) = 2\pi u \frac{(-1)^{u+1} \sin\left(n\left(\theta_j + \frac{\beta}{2}\right)\right) + \sin\left(n\left(\theta_j - \frac{\beta}{2}\right)\right)}{\pi^2 u^2 - \beta^2 n^2} \tag{A14}$$

$$\varepsilon_c(n,u,j) = 2\pi u \frac{(-1)^{u+1} \cos\left(n\left(\theta_j + \frac{\beta}{2}\right)\right) + \cos\left(n\left(\theta_j - \frac{\beta}{2}\right)\right)}{\pi^2 u^2 - \beta^2 n^2} \tag{A15}$$

$$\eta_s(n,u,j) = \frac{\beta^2 n}{\pi} \frac{(-1)^{u} \cos\left(n\left(\theta_j + \frac{\beta}{2}\right)\right) - \cos\left(n\left(\theta_j - \frac{\beta}{2}\right)\right)}{\pi^2 u^2 - \beta^2 n^2} \tag{A16}$$

$$\eta_c(n,u,j) = \frac{\beta^2 n}{\pi} \frac{(-1)^{u+1} \sin\left(n\left(\theta_j + \frac{\beta}{2}\right)\right) + \sin\left(n\left(\theta_j - \frac{\beta}{2}\right)\right)}{\pi^2 u^2 - \beta^2 n^2} \tag{A17}$$

For $\pi u = \beta n$:

$$\varepsilon_s(n,u,j) = \cos\left(n\left(\theta_j - \frac{\beta}{2}\right)\right) - \frac{\sin\left(n\left(\theta_j + \frac{3\beta}{2}\right)\right) - \sin\left(n\left(\theta_j - \frac{\beta}{2}\right)\right)}{2n\beta} \tag{A18}$$

$$\varepsilon_c(n,u,j) = -\sin\left(n\left(\theta_j - \frac{\beta}{2}\right)\right) - \frac{\cos\left(n\left(\theta_j + \frac{3\beta}{2}\right)\right) - \cos\left(n\left(\theta_j - \frac{\beta}{2}\right)\right)}{2n\beta} \tag{A19}$$

$$\eta_s(n,u,j) = \frac{\sin\left(n\left(\theta_j - \frac{\beta}{2}\right)\right)}{2\pi/\beta} - \frac{\cos\left(n\left(\theta_j + \frac{3\beta}{2}\right)\right) - \cos\left(n\left(\theta_j - \frac{\beta}{2}\right)\right)}{4n\pi} \tag{A20}$$

$$\eta_c(n,u,j) = \frac{\cos\left(n\left(\theta_j - \frac{\beta}{2}\right)\right)}{2\pi/\beta} + \frac{\sin\left(n\left(\theta_j + \frac{3\beta}{2}\right)\right) - \sin\left(n\left(\theta_j - \frac{\beta}{2}\right)\right)}{4n\pi} \tag{A21}$$

For $\delta u \neq \beta v$:

$$\gamma_s(u,v) = \frac{2\delta^2 u}{\pi} \frac{(-1)^{u+1} \sin\left(\frac{\pi v}{2\delta}(\delta + \beta)\right) + \sin\left(\frac{\pi v}{2\delta}(\delta - \beta)\right)}{\delta^2 u^2 - \beta^2 v^2} \tag{A22}$$

$$\gamma_c(u,v) = \frac{2\beta^2 v}{\pi} \frac{(-1)^{u+1} \sin\left(\frac{\pi v}{2\delta}(\delta + \beta)\right) + \sin\left(\frac{\pi v}{2\delta}(\delta - \beta)\right)}{\delta^2 u^2 - \beta^2 v^2} \tag{A23}$$

For $\delta u = \beta v$:

$$\gamma_s(u,v) = \frac{2\pi u \cos\left(\frac{\pi}{2}(u - v)\right) - \sin\left(\frac{\pi}{2}(3u + v)\right) - \sin\left(\frac{\pi}{2}(u - v)\right)}{2\pi u} \tag{A24}$$

$$\gamma_c(u,v) = \frac{2\pi u \cos\left(\frac{\pi}{2}(u - v)\right) + \sin\left(\frac{\pi}{2}(3u + v)\right) + \sin\left(\frac{\pi}{2}(u - v)\right)}{2\pi v} \tag{A25}$$

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
