# Peer review of "Exact Two-Dimensional Analytical Calculations for Magnetic Field, Electromagnetic Torque, UMF, Back-EMF, and Inductance of Outer Rotor Surface Inset Permanent Magnet Machines"

_mca, doi:10.3390/mca24010024_

Round 1

Reviewer 1 Report

1. Line 20 : I’m not sure that analytical models are more accurate in results than numerical models ! In times consuming I agree, but in accuracy is not the case.

2.     Figures 2 to 5 must be improved in quality (as Figures in Table 3).

3.     The paper treats the subdomains, so i would like to suggest some text and references to improve the paper:

3.1.   Origin of the method :

Authors of [A] claimed that the first work on the semi-analytical subdomain technique was published by Dubas et al. [13].

3.2.   Contribution in improvement of the method :

New techniques to account for finite soft-magnetic permeabilities have been recently developed, i.e., in the multi-layer model using the Cauchy’s product theorem [B], and in the subdomain technique by applying the superposition principle in both directions [36,37].

An overview of the analytical models in the MaxwellFourier method with a global or local saturation effect has been realized in [37]. According to [C], the Dubas superposition technique [36,37] is very interesting since it enables the magnetic field calculation in the material of slotted geometries. This superposition technique has been implemented in radial-flux electrical machines with(out) PMs supplied by a direct or alternate current [D,E]. 

          3.3.   Some examples of application :

This technique has been extended to : (i) the thermal modeling for steady-state temperature distribution in rotating electrical machines [F], and (ii) elementary subdomains in the rotor and stator regions for full prediction of magnetic field in rotating electrical machines [G].

[A] Pfister, P.; Yin, X.; Fang, Y. Slotted Permanent-Magnet Machines: General Analytical Model of Magnetic Fields, Torque, Eddy Currents, and Permanent-Magnet Power Losses Including the Diffusion Effect. IEEE Trans. Magn. 2016, 52, 1–13.

[13] Dubas, F.; Espanet, C. Analytical Solution of the Magnetic Field in Permanent-Magnet Motors Taking Into Account Slotting Effect: No-Load Vector Potential and Flux Density Calculation. IEEE Trans. Magn. 2009, 45, 2097–2109.

[B] Sprangers, R.L.J.; Paulides, J.J.H.; Gysen, B.L.J.; Lomonova, E.A. Magnetic Saturation in Semi-Analytical Harmonic Modeling for Electric Machine Analysis. IEEE Trans. Magn. 2016, 52, 1–10.

[36] Dubas, F.; Boughrara, K. New Scientific Contribution on the 2-D Subdomain Technique in Polar Coordinates: Taking into Account of Iron Parts. Math. Comput. Appl. 2017, 22, 42.

[37] Dubas, F.; Boughrara, K. New Scientific Contribution on the 2-D Subdomain Technique in Cartesian Coordinates: Taking into Account of Iron Parts. Math. Comput. Appl. 2017, 22, 17.

[C] Hannon, B.; Sergeant, P.; Dupré, L. Two-dimensional Fourier-based modeling of electric machines. In Proceedings of the 2017 IEEE International Electric Machines and Drives Conference, Miami, FL, USA, 21–24 May 2017; pp. 1–8.

[D] Roubache, L.; Boughrara, K.; Dubas, F.; Ibtiouen, R. New Subdomain Technique for Electromagnetic Performances Calculation in Radial-Flux Electrical Machines Considering Finite Soft-Magnetic Material Permeability. IEEE Trans. Magn. 2018, 54, 1–15.

[E] Ben Yahia,M.; Boughrara, K.; Dubas, F.; Roubache, L.; Ibtiouen, R. Two-Dimensional Exact Subdomain Technique of Switched ReluctanceMachines with Sinusoidal Current Excitation. Math. Comput. Appl. 2018, 23, 59.

[F] Boughrara, K.; Dubas, F.; Ibtiouen, R. 2-D Exact Analytical Method for Steady-State Heat Transfer Prediction in Rotating Electrical Machines. IEEE Trans. Magn. 2018, 54, 1–19.

[G] Roubache, L.; Boughrara, K.; Dubas, F.; Ibtiouen, R. Elementary subdomain technique for magnetic field calculation in rotating electrical machines with local saturation effect. Int. J. Comput. Math. Electr. Electron. Eng. 2018, doi:10.1108/COMPEL-11-2017-0481.

Author Response

The authors would like to thank the reviewers for their constructive and pertinent comments/suggestions. Below is the list of actions to address the comments/suggestions.

1. Line 20: I’m not sure that analytical models are more accurate in results than numerical models! In times consuming I agree, but in accuracy is not the case.

Thanks for your comment. Yes, it is completely true. In order to avoid miss understanding, we add some words in page 2. it has been indicated that among analytical models, 2-D and 3-D models are the most accurate models which has been highlighted by yellow color.

2.     Figures 2 to 5 must be improved in quality (as Figures in Table 3).

All the mentioned figures have been modified.

3. The paper treats the subdomains, so i would like to suggest some text and references to improve the paper:

3.1.   Origin of the method:

Authors of [A] claimed that the first work on the semi-analytical subdomain technique was published by Dubas et al. [13].

3.2.   Contribution in improvement of the method :

New techniques to account for finite soft-magnetic permeabilities have been recently developed, i.e., in the multi-layer model using the Cauchy’s product theorem [B], and in the subdomain technique by applying the superposition principle in both directions [36,37].

An overview of the analytical models in the Maxwell‒Fourier method with a global or local saturation effect has been realized in [37]. According to [C], the Dubas’ superposition technique [36,37] is very interesting since it enables the magnetic field calculation in the material of slotted geometries. This superposition technique has been implemented in radial-flux electrical machines with(out) PMs supplied by a direct or alternate current [D, E]. 

          3.3.   Some examples of application:

This technique has been extended to: (i) the thermal modeling for steady-state temperature distribution in rotating electrical machines [F], and (ii) elementary subdomains in the rotor and stator regions for full prediction of magnetic field in rotating electrical machines [G].

[A] Pfister, P.; Yin, X.; Fang, Y. Slotted Permanent-Magnet Machines: General Analytical Model of Magnetic Fields, Torque, Eddy Currents, and Permanent-Magnet Power Losses Including the Diffusion Effect. IEEE Trans. Magn. 2016, 52, 1–13.

 This paper has been added to the references.

[13] Dubas, F.; Espanet, C. Analytical Solution of the Magnetic Field in Permanent-Magnet Motors Taking into Account Slotting Effect: No-Load Vector Potential and Flux Density Calculation. IEEE Trans. Magn. 2009, 45, 2097–2109.

[B] Sprangers, R.L.J.; Paulides, J.J.H.; Gysen, B.L.J.; Lomonova, E.A. Magnetic Saturation in Semi-Analytical Harmonic Modeling for Electric Machine Analysis. IEEE Trans. Magn. 2016, 52, 1–10.

  This paper has been added to the references.

[36] Dubas, F.; Boughrara, K. New Scientific Contribution on the 2-D Subdomain Technique in Polar Coordinates: Taking into Account of Iron Parts. Math. Comput. Appl. 2017, 22, 42.

[37] Dubas, F.; Boughrara, K. New Scientific Contribution on the 2-D Subdomain Technique in Cartesian Coordinates: Taking into Account of Iron Parts. Math. Comput. Appl. 2017, 22, 17.

[C] Hannon, B.; Sergeant, P.; Dupré, L. Two-dimensional Fourier-based modeling of electric machines. In Proceedings of the 2017 IEEE International Electric Machines and Drives Conference, Miami, FL, USA, 21–24 May 2017; pp. 1–8.

  This paper has been added to the references.

[D] Roubache, L.; Boughrara, K.; Dubas, F.; Ibtiouen, R. New Subdomain Technique for Electromagnetic Performances Calculation in Radial-Flux Electrical Machines Considering Finite Soft-Magnetic Material Permeability. IEEE Trans. Magn. 2018, 54, 1–15.

  This paper has been added to the references.

[E] Ben Yahia,M.; Boughrara, K.; Dubas, F.; Roubache, L.; Ibtiouen, R. Two-Dimensional Exact Subdomain Technique of Switched ReluctanceMachines with Sinusoidal Current Excitation. Math. Comput. Appl. 2018, 23, 59.

  This paper has been added to the references.

[F] Boughrara, K.; Dubas, F.; Ibtiouen, R. 2-D Exact Analytical Method for Steady-State Heat Transfer Prediction in Rotating Electrical Machines. IEEE Trans. Magn. 2018, 54, 1–19.

   This paper has been added to the references.

[G] Roubache, L.; Boughrara, K.; Dubas, F.; Ibtiouen, R. Elementary subdomain technique for magnetic field calculation in rotating electrical machines with local saturation effect. Int. J. Comput. Math. Electr. Electron. Eng. 2018, doi:10.1108/COMPEL-11-2017-0481.

  This paper has been added to the references.

Reviewer 2 Report

The paper presents an analytical model allowing the prediction and calculation of the magnetic field as well as the electromagnetic performances of outer rotor surface inset permanent magnet machines. This paper is well presented. However, there are a number of aspects that have to be revised:

1) Recently, a new technique have been developed to consider the iron permeability by harmonic modeling technique [1] or sub-domain technique [2]. Why you did not use one of this techniques to get good results by considering iron permeability.

[1] ''Semi-Analytical Magnetic Field Predicting in Many Structures of Permanent-Magnet Synchronous Machines Considering the Iron Permeability,'' IEEE Trans. Magn., vol. 53, no. 7, July. 2018, Art. no. 8103921 , doi:  10.1109/TMAG.2018.2824278.

[2] New Subdomain Technique for Electromagnetic Performances Calculation in Radial-Flux Electrical Machines Considering Finite Soft-Magnetic Material Permeability,'' IEEE Trans. Magn., vol. 54, no. 4, April. 2018, Art. no. 8103315.

2) Added the nomenclature of Tem, Tco and Trel.

3) Give the connecting matrices of the three-phase current and the stator slots in the Appendix.

4) Other comments can be found in the attachment.

Author Response

The authors would like to thank the reviewers for their constructive and pertinent comments/suggestions. Below is the list of actions to address the comments/suggestions.

1) Recently, a new technique have been developed to consider the iron permeability by harmonic modeling technique [1] or sub-domain technique [2]. Why you did not use one of this techniques to get good results by considering iron permeability.

[1] ''Semi-Analytical Magnetic Field Predicting in Many Structures of Permanent-Magnet Synchronous Machines Considering the Iron Permeability,'' IEEE Trans. Magn., vol. 53, no. 7, July. 2018, Art. no. 8103921 , doi:  10.1109/TMAG.2018.2824278.

[2] New Subdomain Technique for Electromagnetic Performances Calculation in Radial-Flux Electrical Machines Considering Finite Soft-Magnetic Material Permeability,'' IEEE Trans. Magn., vol. 54, no. 4, April. 2018, Art. no. 8103315.

The aim of this paper is to obtain the important quantities like electromagnetic torque, unbalanced magnetic force, back EMF and inductances. To obtain the abovementioned quantities, it is required to calculate the air-gap magnetic flux density and the information about the magnetic flux of the iron part is not influential for these quantities. Although the method of knowing the magnetic flux density in the iron parts is very useful and interesting to have information about local saturation, it makes the model more complex.

2) Added the nomenclature of Tem, Tco and Trel.

The nomenclatures have been added.

3) Give the connecting matrices of the three-phase current and the stator slots in the Appendix.

The connection matrix has been added to the appendix.

4) Other comments can be found in the attachment.

4-1- The next reference should be add:

''Semi-Analytical Magnetic Field Predicting in Many Structures of Permanent-Magnet Synchronous Machines Considering the Iron Permeability,'' IEEE Trans. Magn., vol. 53, no. 7, July. 2018, Art. no. 8103921 , doi: 10.1109/TMAG.2018.2824278.

This reference has been added to the paper. (highlighted by turquoise)

4-2- only one reference have been cited here for this type of machine

Add this reference:

“Analytical predictionof magnetic field in parallel double excitation and spoke-typepermanent-magnet machines accounting for tooth-tips and shape ofpolar pieces,” IEEE Trans. Magn., vol. 48, no. 7, pp. 2121–2137, Jul. 2012.

This reference has been added to the paper. (highlighted by turquoise)

4-3- More citation need to be added here.

[1] ''Magnetic Saturation in Semi-Analytical Harmonic Modeling for Electric Machine Analysis,'' IEEE Trans. Magn., vol. 56, no. 2,  Feb. 2016, Art. no. 8100410 , doi: 10.1109/TMAG.2015.2480708.

[2] ''Nonlinear Analytical Prediction of Magnetic Field and Electromagnetic Performances in Switched Reluctance Machines,'' IEEE Trans. Magn., vol. 53, no. 7, July. 2017, Art. no. 8107311, doi:  10.1109/TMAG.2017.2679686.

This reference has been added to the paper. (ref. 38)

[3] ''Analytical Prediction of Iron-Core Losses in Flux-Modulated Permanent-Magnet Synchronous Machines,'' IEEE Trans. Magn., vol. 55, no. 1,  Jan. 2019, Art. no. 8100410, doi:  10.1109/TMAG.2018.2877164.

This reference has been added to the paper. (highlighted by turquoise)

[4] ''Two-Dimensional Exact Subdomain Technique of Switched Reluctance Machines with Sinusoidal Current Excitation,'' Mathematical and Computational Applications, vol. 23, no. 4, p. 59, Oct. 2018, doi: 10.3390/mca23040059.

This reference has been added to the paper. (ref. 45)

4-4-For Fig. 10 should be use different colors here between Analytical and FEM. The same thing for Fig.11 and 12.

The figures have been modified

4-5-The reference form should be corrected.

The format of the references has been modified.

Round 2

Reviewer 2 Report

The authors have revised all the concerns from me carefully. I would like to accept this paper. Thanks.